# Land Use Carbon Emission Measurement and Risk Zoning under the Background of the Carbon Peak: A Case Study of Shandong Province, China

**Jia-Li Zhong** [1], **Wei Qi** [1,*], **Min Dong** [2], **Meng-Han Xu** [1], **Jia-Yu Zhang** [1], **Yi-Xiao Xu** [1] and **Zi-Jie Zhou** [3]

[1]   College of Resource and Environment, Shandong Agricultural University, Tai'an 271018, China
[2]   Xintai Modern Agriculture Development Service Center, Tai'an 271120, China
[3]   Urban Planning&Design Institute of Shenzhen, Shenzhen 518049, China
*   Correspondence: qiwei@sdau.edu.cn

**Abstract:** Land use and cover change (LUCC) has a non-negligible impact on both carbon emissions and carbon sinks. Based on the analysis of land use dynamics in Shandong Province, this study simulates land use changes in Shandong Province in 2030 under the Natural Development Scenario (NDS) and Sustainable Development Scenario (SDS), classifies the risk level of carbon emissions in Shandong Province using the Land Use Carbon Emission (LUCE) risk indexes, and compares the differences between the risk level regions under NDS and SDS. This study shows that under the influence of LUCC, the carbon emissions in Shandong province increased significantly, from 90.5591 million tons in 2000 to 493.538 million tons in 2020, with urban land being the main source of carbon emissions, which increased from 90.0757 million tons in 2000 to 490.139 million tons in 2020. The main source of the increase in urban land was cropland. The LUCE was positively correlated with urban land area. The LUCE of SDS was 7.2423 million tons less than that of NDS. From 2000 to 2020, the risk areas of LUCE in Shandong province were mainly no-risk and mild-risk areas. The number of moderate-risk areas and high-risk areas of SDS was less than that of NDS. The rational organization of land use is important for Shandong Province to achieve low-carbon development.

**Keywords:** PLUS model; gray forecasting model; land use dynamic degree; risk index

## 1. Introduction

As mankind entered the industrial age, carbon dioxide emissions began to show explosive growth, which led to the aggravation of global warming and sea level rise. Carbon emissions have become a severe problem and are a challenge faced by the international community [1]. Achieving carbon peak and carbon neutrality has become a worldwide extensive and profound social and economic transformation. Governments of global countries have formulated a series of policies and laws for the development of renewable energy to restrain greenhouse gas emissions [2]. In China, carbon peaking and carbon neutrality have become new development directions, and low-carbon development will become the mainstream of future development [3]. Land use and cover change (LUCC) is both a source of atmospheric greenhouse gases and an important carbon sink. It plays an important role in regulating the climate system. Therefore, addressing land use issues can help combat climate change.

Based on the above background, scholars have studied Land Use Carbon Emissions (LUCE) from different professional perspectives. Hong et al. [4] comprehensively evaluated global LUCE from 1961 to 2017 and revealed the drivers of LUCE. Based on the land use classification of Landsat Image Data, Cui et al. [5] calculated the LUCE of the Yangtze River Delta urban agglomeration by using the LUCE coefficient. Scott et al. [6] found that land use changes and anthropogenic changes in land structure would promote the release of more carbon dioxide. For the calculation of carbon emissions, scholars have used a variety

of model methods to calculate carbon emissions. Cong et al. [7] predicted China's carbon emission scenario in 2030 by using the BP neural network model. Hu et al. [8] studied the carbon emission effect of land use in Heilongjiang Province by using the data of land use and energy statistics through the coefficient accounting method. Xu et al. [9] integrated multi-objective planning (MOP) and a system dynamics (SD) simulation model to optimize land use structure in Hubei Province, reducing carbon emissions and improving land use efficiency. Li et al. [10] constructed an evaluation index system for the low-carbon intensive utilization of urban land through Analytic Network Process (ANP), and evaluated the level of land use in Nanjing. Ren et al. [11] combined the STIRPAT model with the CSO-FLN model to predict the carbon emissions in Guangdong Province. Zhu et al. [12] predicted the peak year of carbon emissions in Shanxi Province by building an IPAT model. Zhang et al. [13] used a nonparametric additive regression model to study the impact of renewable energy investment on China's carbon emissions. Long et al. [14] calculated the carbon emissions of Changxing County from four aspects: energy, industrial process, product utilization, agriculture, forestry, and other land use wastes. Zhu et al. [15] used partial least squares regression to study the pattern and driving factors of carbon emissions in rural–urban fringe communities in Southeast Asia. Wei et al. [16] used the gray forecasting model to predict the carbon emissions of Guizhou Province in 2030.

The research on LUCC prediction is also being developed in-depth, and the main models often used by scholars are the FLUS model [17] and the CLUE-S model [18]. The latest LUCC prediction model is the PLUS model developed by Liang et al. [19]. Lou et al. [20] used the PLUS model to predict the LUCC in the Yellow River Basin and estimate its ecological service value. Gao et al. [21] used the PLUS model to simulate the LUCC of Nanjing in 2025 under four scenarios: Business As Usual (BAU), Rapid Economic Development (RED), Ecological Land Protection (ELP), and Ecological and Economic Balance (EEB) and evaluated ecological risks under four scenarios. Wang et al. [22] compared the FLUS model with the PLUS model and showed that the PLUS model has a higher accuracy. The PLUS model can simulate the future LUCC under different scenarios and can compare the pros and cons of different scenarios. Compared with other LUCC prediction models, the PLUS model has more advantages.

In summary, few studies have focused on the impact of future LUCC on LUCE. Regarding the projection of carbon emissions, most studies can only be based on fixed scenarios, ignoring the impact of LUCC on carbon emissions under different scenarios. In terms of research methods, the gray forecasting model [16] is widely used to perform research in forecasting because of its low computational effort and high forecasting accuracy, but fewer studies have combined the PLUS model with the gray forecasting model. The combination of the PLUS model and the gray prediction model effectively predicts the impact of LUCC on carbon emissions under different future scenarios. The method effectively fills this research gap and provides recommendations for reducing carbon emissions from the perspective of LUCC.

We used the Land Use Dynamic Degree (LUDD) and Land Use Transfer Matrix (LUTM) to analyze land use change in Shandong Province from 2000 to 2020. Then, we simulated the LUCC in 2030 under NDS and SDS and assessed the LUCE in Shandong Province from 2000 to 2030. Finally, based on the division of LUCE risk zones, the differences in LUCE under NDS and SDS can be of practical significance for policymakers in formulating land use policies.

The paper is structured as follows. Section 2 demonstrates the research framework, data sources, and methodology. This section presents information about the study area. Section 3 analyses the LUCC and LUCE changes from 2000 to 2030 and analyses the LUCE risk in different regions from 2000 to 2030. The differences in LUCE under NDS and SDS in 2030 are compared. Section 4 analyses the strengths of the methodology used in this paper and its limitations. Section 5 summarizes the findings.

## 2. Materials and Methods

### 2.1. Study Area

Shandong Province (114°48′–122°42′ E, 34°23′–38°17′ N) is located on the eastern coast of China (Figure 1), 721.03 km long from east to west, 437.28 km long from north to south, bordering the Yellow Sea in the East, the Bohai Sea and Hebei in the north, Henan in the west, Jiangsu and Anhui in the south, and is rich in sea and land resources. Shandong Province has a complex landform, with a land area of about 155,800 km², which can be roughly divided into plains, platforms, hills, mountains, and other landform types. The climate belongs to the temperate monsoon climate, with the same period of rain and heat, and an annual average temperature of 11–14 °C. Shandong Province is one of the provinces with the fastest economic development and the largest energy consumption in China. The energy consumption structure is dominated by coal and oil. While the regional GDP has increased year by year, the total carbon emissions have also increased year by year. In 2021, Shandong's GDP was 830,959 billion yuan, ranking third among China's provinces. Although the carbon emission intensity of Shandong Province has gradually declined in recent years, there are still problems that cannot be ignored [23]. Under the background of China's strategy of achieving a carbon peak by 2030, Shandong Province is facing enormous pressure to reduce carbon. However, there are few studies on carbon emission prediction and regional carbon emission risk warnings. It is of great significance to develop a low-carbon sustainable development strategy to predict carbon emissions and identify risk levels in different regions by using multiple models.

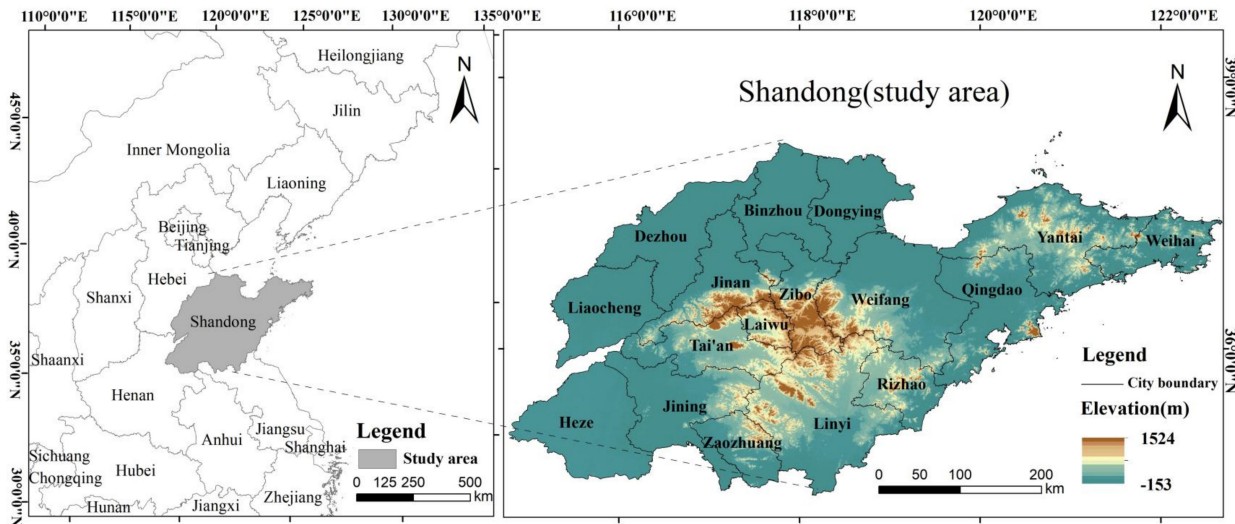

**Figure 1.** Map of the study area.

### 2.2. Data Sources

Multivariate data were used in this study, as shown in Table 1. The land use data were sourced from Google Earth Engine (GEE) with a spatial resolution of 30 m. The data of the Digital Elevation Model (DEM) were sourced from the NASA website with a spatial resolution of 30 m. The GDP data came from the Resource Science Data Center of the Chinese Academy of Sciences with a spatial resolution of 1 km. Data on population, temperature, and rainfall were obtained from the National Earth System Science Data Center with a spatial resolution of 1 km. Railway, highway, and primary road data were sourced from BigMap. The energy consumption data came from the China Energy Statistical Yearbook and the Shandong Statistical Yearbook. The above raster data and vector data were preprocessed and unified into raster data with a spatial resolution of 150 m.

**Table 1.** The main data and data sources.

| Data | Year (s) | Spatial Resolution | Sources |
|---|---|---|---|
| Land use data | 2000–2020 | 30 m | http://doi.org/10.5281/zenodo.4417809 (accessed on 5 April 2022) |
| DEM | 2020 | 30 m | https://lpdaac.usgs.gov/ (accessed on 20 March 2022) |
| GDP | 2019 | 1000 m | https://www.resdc.cn/ (accessed on 25 March 2022) |
| Population | | | |
| Annual average temperature | 2020 | 1000 m | http://www.geodata.cn/ (accessed on 18 April 2022) |
| Annual average precipitation | | | |
| Proximity to highway | | | |
| Proximity to railway | 2020 | | http://www.bigemap.com/ (accessed on 6 April 2022) |
| Proximity to primary road | | | |
| China Energy Statistical Yearbook | 2000–2020 | | http://www.stats.gov.cn/ (accessed on 29 April 2022) |
| Shandong Statistical Yearbook | 2000–2020 | | http://tjj.shandong.gov.cn/ (accessed on 28 April 2022) |

*2.3. Methodological Framework*

We used LUDD to analyze LUCC in Shandong Province from 2000 to 2020. For the calculation of LUCE, we adopted a combination of the direct estimation method and the indirect estimation method. For cropland, forestland, grassland, water area, and bare land, the direct estimation methods were used to calculate their LUCE [24]. The consumption of fossil energy is the main source of carbon dioxide, and the human use of fossil energy is mainly in urban land, so the indirect measurement method was used to calculate the LUCE of urban land. Regarding the prediction of LUCE, this study used the PLUS model to simulate LUCC in 2030 under two different scenarios. For cropland, forestland, grassland, water, and bare land, the direct estimation method was used to calculate their LUCEs in 2030 [19]. When using the indirect estimation method to calculate the LUCE of urban land in 2030, the gray forecasting model was used to predict the carbon emission coefficient of urban land in 2030 [25]. Finally, according to the LUCE from 2000 to 2030, the LUCE risk indexes from 2000 to 2030 were calculated, the natural breakpoint method was used to classify the LUCE risk indexes in different regions, and the regional risk levels were compared in two different scenarios. The two different scenarios can represent two different regional development policies adopted by policymakers. The research framework of this study is shown in Figure 2.

2.3.1. Analysis of Land Use Quantity Change

We analyzed the quantitative changes in land use types through the LUDD. The LUDD plays an important role in quantitatively describing land use change, predicting future land use change trends, and comparing differences in different land uses. The calculation formula is as follows [26]:

$$X = \frac{X_b - X_a}{X_a} \times \frac{1}{T} \times 100 \tag{1}$$

where $X_a$ represents the area of a certain land use type in the starting year, $X_b$ represents the area of a certain land use type in the last year, $T$ represents the research period and unit is a year, and $X$ represents the LUDD of a certain land use type in $T$ year.

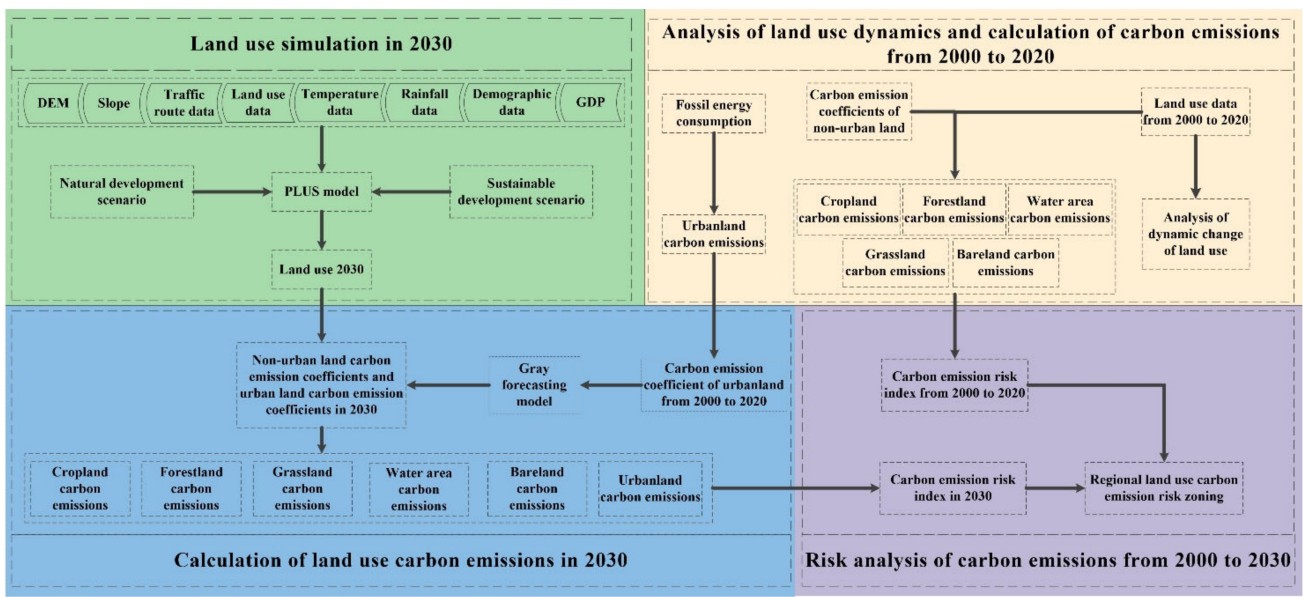

**Figure 2.** The research framework of this study.

2.3.2. Analysis of Land Use Structure Change

The LUTM is derived from systematic analysis. It is widely used to analyze the quantitative description of system state and state transition. The LUTM can comprehensively and specifically analyze the quantitative transition of land use types and changes in land use structure. Its formula is as follows [27]:

$$S_{ij} = \begin{bmatrix} S_{11}S_{12}\cdots\cdots S_{1n} \\ S_{21}S_{22}\cdots\cdots S_{2n} \\ \cdots\cdots\cdots\cdots \\ S_{n1}S_{n2}\cdots\cdots S_{nn} \end{bmatrix} \tag{2}$$

where $S$ is the land use area, $i$ represents the land use type at the beginning of the study period, $j$ represents the land use type at the end of the study period, and $n$ is the number of land use types.

2.3.3. LUCE Calculation

LUCE can be divided into direct carbon emissions and indirect carbon emissions [28]. Direct carbon emissions mainly refer to the carbon emissions caused by the replacement of ecosystem types caused by LUCC, such as deforestation, reclamation of lakes, farmland cultivation, and grassland degradation. Indirect carbon emissions mainly refer to all anthropogenic carbon emissions carried by various land use types, including heating in residential areas, exhaust gas from transportation land, and so on. We focused on calculating the carbon emissions and carbon absorption of the main land use types caused by human activities to obtain the net carbon emissions of different land use types. Emission is positive and absorption is negative.

The direct estimation method was used to calculate the carbon emissions of cropland, forestland, grassland, water area, and bare land. The formula is as follows [24]:

$$E = \sum e_i = \sum S_i \times U_i \tag{3}$$

where $E$ is the total LUCEs, $e_i$ is the carbon emissions generated by land use types, $S_i$ is the area of land use types, $U_i$ is the carbon emission coefficient per unit area of land use types, and $i$ is different land use types.

According to Liu et al. [24], Wu [29], and IPCC research results, it was determined that the carbon emission coefficient of cropland in this study was 0.0372 kg(C)/(m²·a),

and the carbon emission coefficient of forestland was $-0.0487$ kg(C)/(m²·a), the carbon emission coefficient of grassland was $-0.0191$ kg(C)/(m²·a), the carbon emission coefficient of water area was $-0.0253$ kg(C)/(m²·a), and the carbon emission coefficient of bare land was $-0.0005$ kg(C)/(m²·a). Affected by human activities, the carbon emission coefficient of urban land does not have a fixed value. At present, the main source of carbon dioxide is the consumption of fossil energy, and urban land is the place with the most fossil energy consumption, so we used fossil energy consumption data to represent the carbon emission of urban land. The formula is as follows [30]:

$$E_a = \sum e_{ni} = \sum E_{ni} \times V_i \times f_i \tag{4}$$

$E_a$ is the LUCE of urban land, $E_{ni}$ is the consumption of fossil energy, $V_i$ is the coefficient of converting fossil energy into standard coal, $f_i$ is the carbon emission coefficient of fossil energy, and $e_{ni}$ is the carbon emission of fossil energy.

### 2.3.4. Gray Forecasting Model

The gray forecasting model is a mathematical statistical prediction model that predicts unknown information through known information [31]. Because the fossil energy consumption data in 2030 cannot be obtained, we used the gray forecasting model GM (1,1) to predict the carbon emission coefficient of urban land in 2030 based on obtaining the carbon emission coefficient of urban land from 2000 to 2020. The formula is as follows [25].

Set the original sequence as:

$$x^{(0)} = [x^{(0)}(1), x^{(0)}(2)\ldots\ldots x^{(0)}(n)]\left(x^{(0)}(i) \geq 0, i = 1, 2, 3, \ldots\ldots, n\right) \tag{5}$$

Generate the accumulation sequence $x^{(1)}$:

$$x^{(1)} = [x^{(1)}(1), x^{(1)}(2), \ldots\ldots, x^{(1)}(n)] \tag{6}$$

When $x^{(0)}(1) = x^{(1)}(1)$,

$$x^{(1)}(t) = \sum_{i=1}^{t} x^{(0)}(i) = x^{(1)}(t-1) + x^{(0)}(t), t = 2, 3, \ldots\ldots, n \tag{7}$$

Then the differential equation of GM (1,1) model is:

$$\frac{dx^{(1)}}{dt} + ax^{(1)} = \mu \tag{8}$$

where, $a$ is the development coefficient, reflecting the development trend of $x^{(1)}$ and the original sequence $x^{(0)}$. $\mu$ is called the endogenous control coefficient, which reflects the change relationship between data.

Let us assume:

$$\hat{\alpha} = (a, \mu)^T \tag{9}$$

Then the least square method is used to obtain:

$$\hat{\alpha} = (B^T B)^{-1} B^T Y_1 \tag{10}$$

In Formula (10):

$$B = \begin{pmatrix} -\frac{1}{2}\left(x^{(1)}(1) + x^{(1)}(2)\right) & 1 \\ -\frac{1}{2}\left(x^{(1)}(2) + x^{(1)}(3)\right) & 1 \\ \vdots & \vdots \\ -\frac{1}{2}\left(x^{(1)}(n-1) + x^{(1)}(n)\right) & 1 \end{pmatrix} \tag{11}$$

$$Y_1 = \begin{pmatrix} x^{(0)}(2) \\ x^{(0)}(3) \\ \vdots \\ x^{(0)}(n) \end{pmatrix} \tag{12}$$

The solution to (8) is obtained as:

$$\hat{x}^{(1)}(t+1) = \left(x^{(0)}(1) - \frac{\mu}{\alpha}\right)e^{-\alpha t} + \frac{\mu}{\alpha} \tag{13}$$

$x^{(1)}$ is an accumulation sequence of $x^{(0)}$, By subtracting the Formula (13), the predicted value of $x^{(0)}$ can be obtained:

$$\hat{x}^{(0)}(t+1) = \hat{x}^{(1)}(t+1) - \hat{x}^{(1)}(t) \tag{14}$$

where, $t$ = 1,2,3, . . . , $n$. Then, the final prediction Formula (15) is obtained:

$$\hat{x}^{(0)}(t+1) = (1 - e^{\alpha})\left(x^{(0)}(1) - \frac{\mu}{\alpha}\right)e^{-\alpha t} \tag{15}$$

2.3.5. PLUS Model

The PLUS model integrates a land expansion analysis strategy and a CA model based on multi-type random patch seeds. It can realize the probability calculation of suitability based on driving factors in the model and the new training of samples based on the conversion of various types of land use between two periods of land use data in the ANN-CA model and use the random forest method to mine various driving factors. Compared with most existing land use simulation models, this model has a higher simulation accuracy and landscape pattern indicators that are closer to the real landscape. The PLUS model can help decision-makers better manage future land use dynamics and better achieve sustainable development goals [19]. Therefore, we used the PLUS model to simulate and predict future land use scenarios in the study area.

Based on the land use data in 2010 and 2020, we selected DEM, slope, proximity to the highway, proximity to the railway, proximity to the primary road, GDP, population, annual average precipitation, and annual average temperature (Figure 3). The 9 driving factors of LUCC and the LUCC map in 2010 were incorporated into the PLUS model for calculation, and finally, the simulated LUCC map in 2020 was obtained. The simulated 2020 LUCC map was compared with the real 2020 LUCC map, and the simulation results were checked using the Kappa coefficient and overall accuracy to ensure the accuracy of the results. Finally, by setting the NDS and SDS, the LUCC in 2030 was predicted and the differences between the two scenarios were compared.

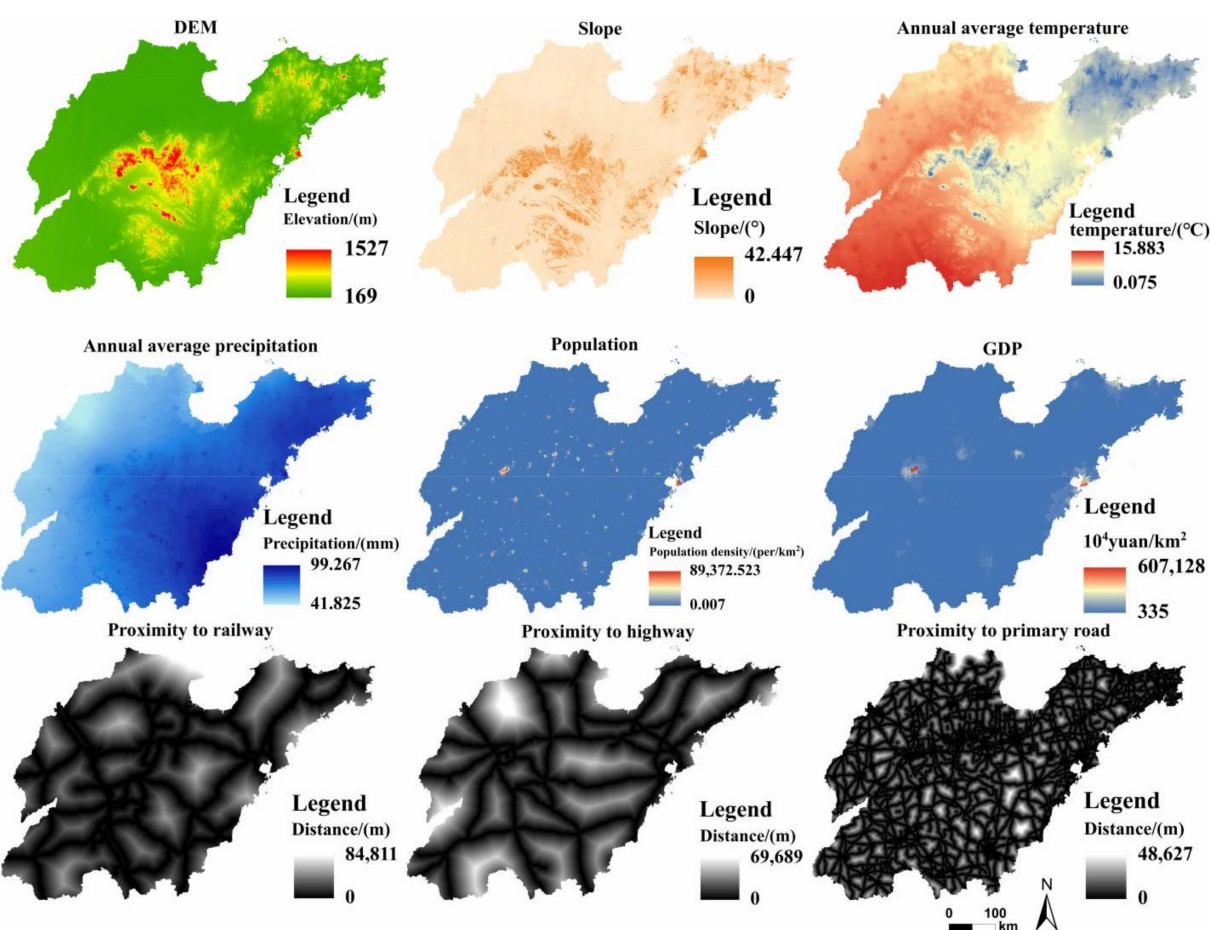

**Figure 3.** Driving factors affecting LUCC.

2.3.6. LUCE Risk Indexes Calculation

LUCE will have a great impact on the local ecosystem, which will lead to the occurrence of ecological risks. A higher LUCE risk indexes indicate more carbon emissions in the region. Therefore, we used the LUCE risk indexes to measure the local ecological risk. The larger the LUCE risk index, the higher the local ecological risk, and the lower the LUCE risk index, the lower the local ecological risk. The formula is as follows [32]:

$$C_{R_i} = \sum_{i}^{j} \frac{S_{ji} P_{ji}}{S} \tag{16}$$

where $C_{R_i}$ is the LUCE risk index of the $i$ county in Shandong Province. $S_{ji}$ is the land use area of class $j$ in County $i$. $P_{ji}$ is the carbon emission coefficient of class $j$ land use type in County $i$. $S$ is the total area of the study area.

## 3. Results

### 3.1. Temporal and Spatial Dynamic Changes of Land Use

3.1.1. Land Quantity Change

Table 2 shows the changes in the number of various types of land in Shandong Province from 2000 to 2020. The cropland area has shown a downward trend in the past 20 years, with a total reduction of 11,459.32 km². The forestland area showed a trend of decline first and then increase, with an overall increase of 1075.61 km². The grassland area decreased by 1525.82 km². The water area increased by 2094.08 km² in total. The urban land area has shown an increasing trend in the past 20 years, and the overall area increased by

11,616.39 km². The LUDD is in the order of water area > urban land > forestland > cropland > grassland > bare land.

**Table 2.** Change of land use types in Shandong Province from 2000 to 2020 (Unit of area change: km², unit of LUDD: %).

| Land Use Type | 2000–2010 | | 2010–2020 | | 2000–2020 | |
|---|---|---|---|---|---|---|
| | Area Change | LUDD | Area Change | LUDD | Area Change | LUDD |
| Cropland | −5804.08 | −0.49 | −5655.24 | −0.50 | −11,459.32 | −0.97 |
| Forestland | −106.34 | −0.16 | 1181.95 | 1.78 | 1075.61 | 1.59 |
| Grassland | −630.81 | −1.72 | −895.01 | −2.95 | −1525.82 | −4.16 |
| Water area | 1662.14 | 5.02 | 431.93 | 0.87 | 2094.08 | 6.32 |
| Bare land | −413.10 | −1.87 | −1387.85 | −7.71 | −1800.95 | −8.14 |
| Urban land | 5292.18 | 2.31 | 6324.21 | 2.24 | 11,616.39 | 5.08 |

3.1.2. Land Structure Change

According to the land use structure chart (Figure 4), the proportion of cultivated land in the total land area decreased year by year, and the proportion of urban land increased year by year. The proportion of forestland, grassland, water area, and bare land in the total land area changed slightly compared with cropland and urban land.

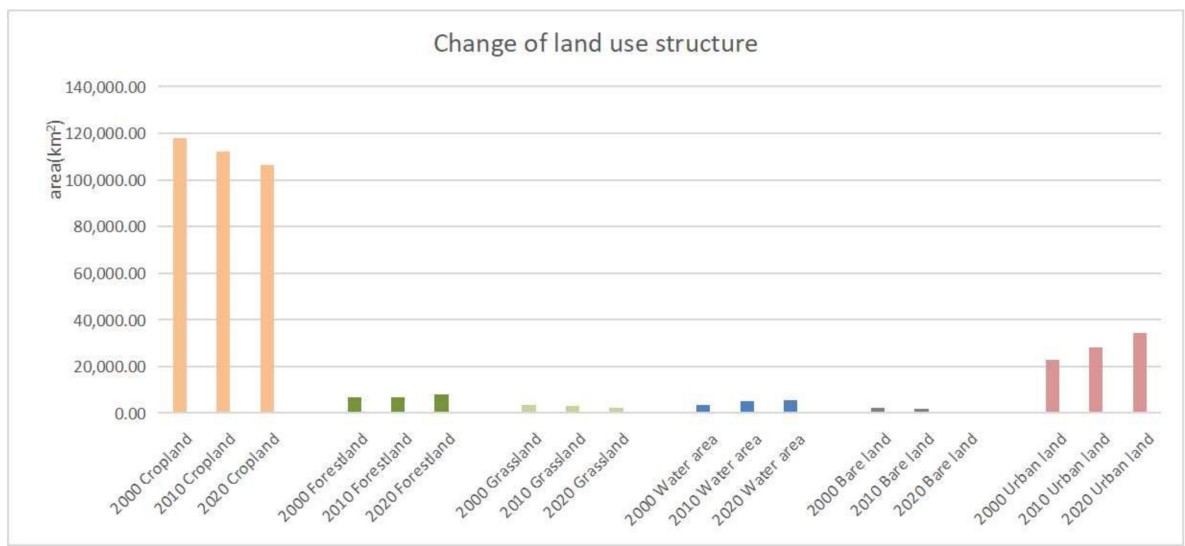

**Figure 4.** Change of land use structure.

Tables 3–5 show the LUTM from 2000 to 2020. From 2000 to 2010, the mainland conversion was from cropland to urban land, accounting for 48.99% of the total land transferred out. Grassland was mainly transferred out as cropland, with an area of 849.76 km², accounting for 70.80% of the total grassland transferred out. The main types of cropland transferred out were urban land and water area, with an area of 5377.34 km² and 1146.31 km², respectively, accounting for 71.51 and 15.24% of the total cropland transferred out. Forestland was mainly transferred out as cropland, with an area of 652.10 km², accounting for 89.98% of the total transferred out area of forestland. The main type of land transferred out of bare land was water area, with an area of 446.87 km², accounting for 73.85% of the total area transferred out of bare land. The main types of land transferred from water areas were cropland, urban land, and bare land, with areas of 127.19 km², 179.84 km², and 109.69 km², respectively, accounting for 30.25%, 42.78%, and 26.09% of the total area transferred from the water area.

**Table 3.** LUTM in Shandong Province from 2000 to 2010 (Unit: km$^2$).

| Land Type | Grassland | Cropland | Urban Land | Forestland | Bare Land | Water Area | Total |
|---|---|---|---|---|---|---|---|
| Grassland | 2468.45 | 849.76 | 119.12 | 166.68 | 52.90 | 11.79 | 3668.69 |
| Cropland | 521.15 | 110,331.77 | 5377.34 | 451.49 | 23.04 | 1146.31 | 117,851.09 |
| Urban land | 0.41 | 23.42 | 22,380.26 | 0.05 | 6.37 | 476.80 | 22,887.29 |
| Forestland | 36.29 | 652.10 | 35.53 | 6020.64 | 0.00 | 0.79 | 6745.34 |
| Bare land | 8.06 | 62.78 | 87.39 | 0.00 | 1607.85 | 446.87 | 2212.94 |
| Water area | 3.53 | 127.19 | 179.84 | 0.16 | 109.69 | 2891.25 | 3311.66 |
| Total | 3037.88 | 112,047.01 | 28,179.47 | 6639.01 | 1799.84 | 4973.81 | 156,677.02 |

**Table 4.** LUTM in Shandong Province from 2010 to 2020 (Unit: km$^2$).

| Land Type | Grassland | Cropland | Urban Land | Forestland | Bare Land | Water Area | Total |
|---|---|---|---|---|---|---|---|
| Grassland | 1621.24 | 897.53 | 76.19 | 418.41 | 7.85 | 16.67 | 3037.88 |
| Cropland | 513.88 | 104,269.77 | 5449.88 | 1245.29 | 4.91 | 563.29 | 112,047.01 |
| Urban land | 0.00 | 15.53 | 27,925.76 | 0.00 | 2.72 | 235.46 | 28,179.47 |
| Forestland | 5.00 | 456.59 | 21.53 | 6155.57 | 0.00 | 0.31 | 6639.01 |
| Bare land | 2.25 | 72.41 | 614.59 | 0.00 | 331.34 | 779.27 | 1799.84 |
| Water area | 0.5 | 679.95 | 415.73 | 1.69 | 65.18 | 3810.74 | 4973.81 |
| Total | 2142.88 | 106,391.77 | 34,503.68 | 7820.96 | 412.00 | 5405.74 | 156,677.02 |

**Table 5.** LUTM in Shandong Province from 2000 to 2020 (Unit: km$^2$).

| Land Type | Grassland | Cropland | Urban Land | Forestland | Bare Land | Water Area | Total |
|---|---|---|---|---|---|---|---|
| Grassland | 1541.36 | 1255.37 | 213.71 | 611.57 | 14.18 | 32.51 | 3668.69 |
| Cropland | 552.98 | 103,915.15 | 10,848.83 | 1244.66 | 31.59 | 1257.89 | 117,851.09 |
| Urban land | 0.13 | 133.16 | 22,244.36 | 0.45 | 10.17 | 499.03 | 22,887.29 |
| Forestland | 46.71 | 670.68 | 64.08 | 5962.95 | 0.11 | 0.81 | 6745.34 |
| Bare land | 1.46 | 105.64 | 717.77 | 0.00 | 329.49 | 1058.58 | 2212.94 |
| Water area | 0.23 | 311.78 | 414.95 | 1.33 | 26.46 | 2556.92 | 3311.66 |
| Total | 2142.88 | 106,391.77 | 34,503.68 | 7820.96 | 412.00 | 5405.74 | 156,677.02 |

During the period from 2010 to 2020, the land was mainly transferred from cropland to urban land, accounting for 43.38% of the transferred out area of all land types. Cropland was the main type of grassland transferred out, with an area of 897.53 km$^2$, accounting for 63.36% of the grassland transferred area. The main type of cropland transferred out was urban land, with an area of 5449.88 km$^2$, accounting for 70.07% of the cropland transferred out. Forestland was mainly converted into cropland, with an area of 456.59 km$^2$, accounting for 94.45% of the total area of forestland transferred out. Bare land was mainly converted to urban land and water area, with an area of 614.59 and 779.27 km$^2$, respectively, accounting for 41.85 and 53.07% of the transferred out area of bare land, respectively. The water area was mainly converted to cropland and urban land, with an area of 679.95 km$^2$ and 415.73 km$^2$, respectively, accounting for 58.46% and 35.74%, respectively.

From 2000 to 2020, land transfer was mainly from cropland to urban land, accounting for 53.90% of the total land transferred out of all land types. Grassland was mainly converted to cropland, with an area of 1255.37 km$^2$, accounting for 59.01% of the total area of grassland. Cropland was mainly converted to urban land, with an area of 10,848.83 km$^2$, accounting for 77.85% of the total area of cropland transferred out. Forest land was mainly converted to cropland, with an area of 670.68 km$^2$, accounting for 85.72% of the total area of forestland transferred out. Bare land was mainly converted to urban land and water area, with an area of 717.77 and 1058.58 km$^2$, respectively, accounting for 38.11 and 56.20% of the transferred out area of bare land, respectively. The water area was mainly converted into cropland and urban land, with an area of 311.78 and 414.95 km$^2$, respectively, accounting for 41.31 and 54.98% of the water area transferred out, respectively.

### 3.1.3. Land Spatial Change

Combining Table 5 with Figure 5, the following results can be analyzed: from 2000 to 2020, the area of other land types converted to forestland was 1858.01 km², and the distribution area was mainly concentrated in the following seven regions, including Weihai, Yantai, Eastern Qingdao, Jinan, Zibo, Laiwu, Tai'an, Rizhao, Linyi, and Zaozhuang. The area of other land types converted to cropland was 2476.62 km², mainly distributed in the following three regions, including Zaozhuang, Jining, and the Yellow River Delta. The area of other land types converted to urban land was 12,259.33 km², which increased in varying degrees throughout Shandong Province, mainly distributed in the main urban areas of various regions. The area of other land types converted to grassland was 601.52 km², which was mainly distributed in the following five areas, including Jining, Linyi, Zaozhuang, Yantai, and Laiwu.

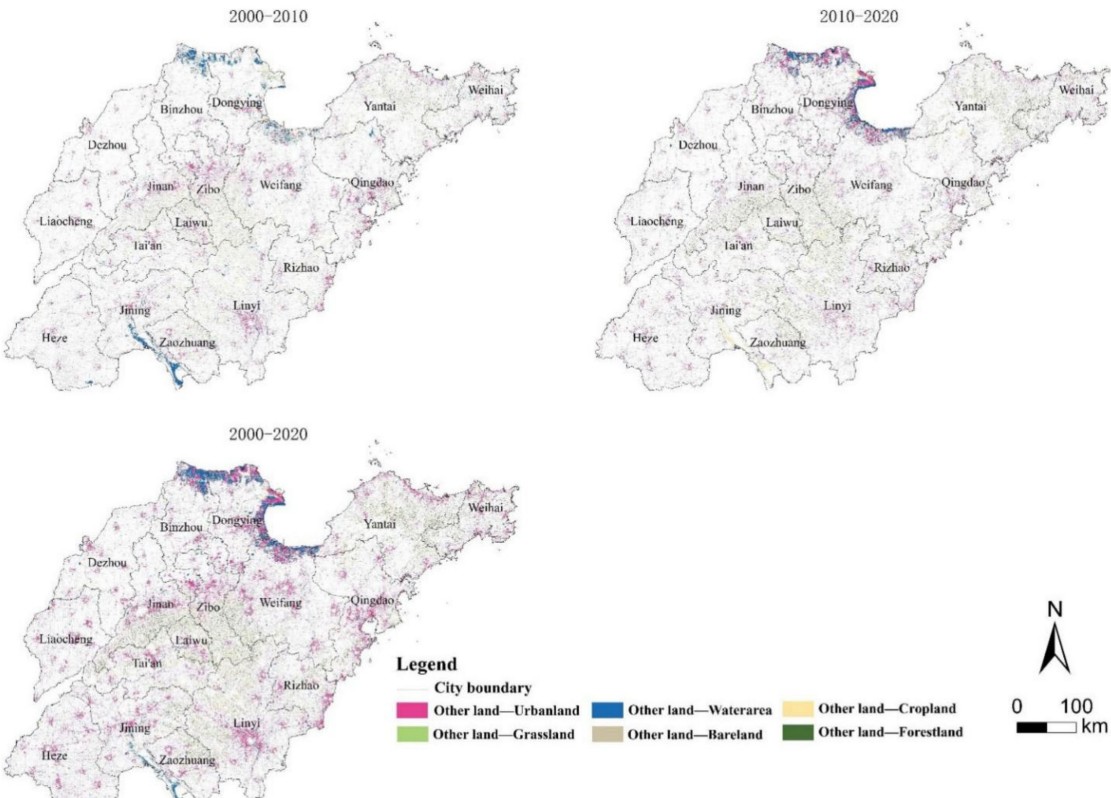

**Figure 5.** Transfer map of land use types from 2000 to 2020.

### 3.2. Land Use Scenario Simulation

Before simulating the 2030 land use, we entered the drivers into the PLUS model and then tested the simulation accuracy of the model, the results of which are shown in Figure 6. Figure 7 shows the land use map (a) under the 2030 natural development scenario (NDS) and the land use map (b) under the 2030 sustainable development scenario (SDS) predicted by the PLUS model. The NDC is based on the rate of LUCC from 2010 to 2020 and does not limit the conversion of land types. The SDC is based on the rate of LUCC from 2010 to 2020, limiting the conversion of forestland, water area, and grassland to other land types, thereby maximizing carbon sinks.

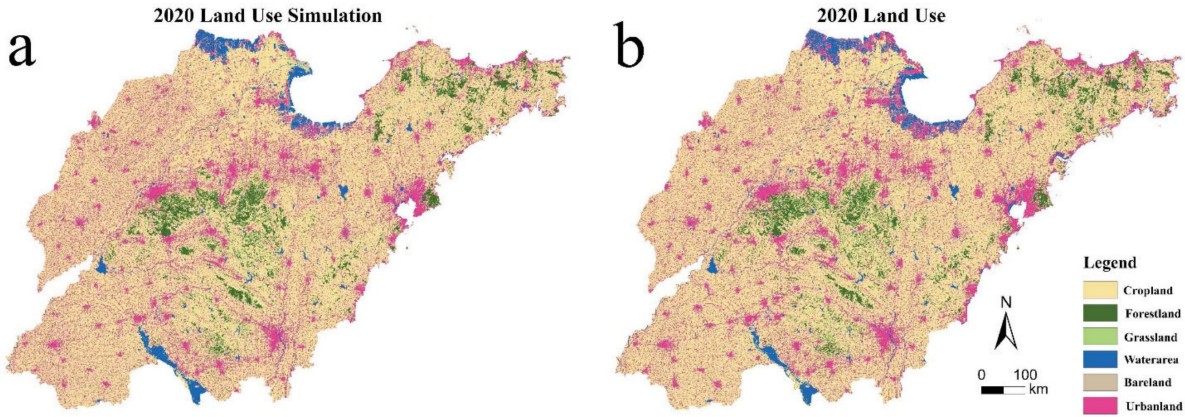

**Figure 6.** (**a**) is the land use simulation map obtained by the PLUS model operation, and (**b**) is the current land use map. After the Kappa coefficient test, the results show that the Kappa coefficient was 0.7740, and the overall accuracy of the PLUS model was 89.12%, which proves that the simulation results of the PLUS model had a high accuracy. It is feasible to use the PLUS model to predict LUCC in Shandong Province in 2030. The 2020 land use map simulated by the PLUS model (**a**) and the real 2020 land use map (**b**).

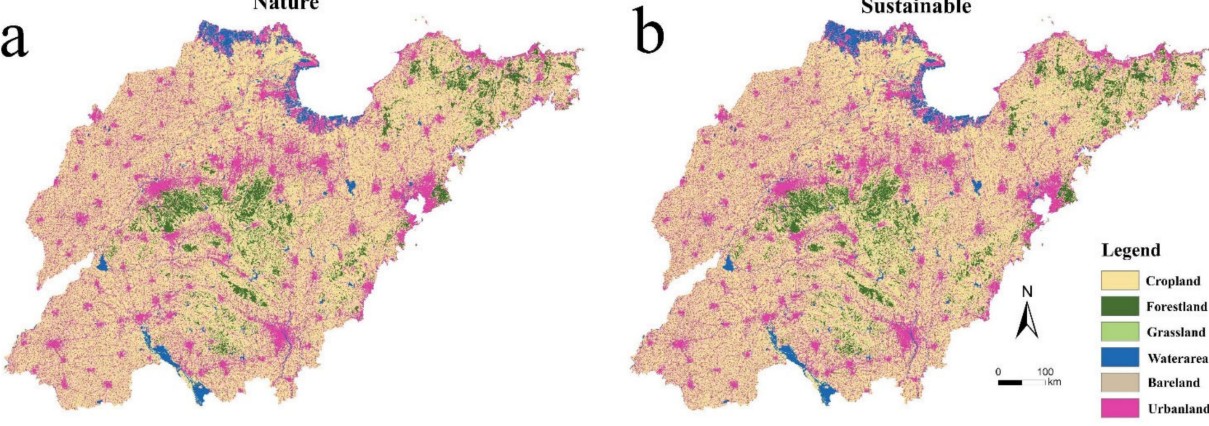

**Figure 7.** The 2030 natural development scenario land use map (**a**) and the 2030 sustainable development scenario land use map (**b**) simulated by the PLUS model.

### 3.3. Land Use Carbon Emissions

We used the gray forecasting model to predict the urban land carbon emission coefficient in 2030. Firstly, based on the energy consumption data from 2000 to 2020, the LUCE coefficient of urban land use in 2000 was 3.9356 kg(C)/(m²·a). The LUCE coefficient of urban land use in 2010 was 12.9676 kg(C)/(m²·a), and that of urban land use in 2020 was 14.2054 kg(C)/(m²·a). Then, it was predicted that the LUCE coefficient of urban land use in 2030 would be 15.5482 kg(C)/(m²·a) by using the gray forecasting model.

According to the analysis in Table 6, from 2000 to 2020, the LUCE in Shandong Province showed an overall growth trend, from 90.5591 million tons in 2000 to 493.538 million tons in 2020, with an average annual increase of 20.1489 million tons. Carbon emissions from urban land increased from 90.0757 million tons in 2000 to 49.0139 million tons in 2020, with an average annual growth of 20.032 million tons. This result shows that the total carbon emissions of Shandong Province from 2000 to 2020 were in direct proportion to the carbon emissions of urban land. The land use carbon sink also showed an increasing trend, from 483,500 tons in 2000 to 558,800 tons in 2020, an increase of 15.57%. The forestland carbon sink increased from 328,500 tons in 2000 to 380,900 tons in 2020, an increase of 15.95%. In 2000, the forestland carbon sink accounted for 67.94% of the total carbon sink, and in 2020,

the forestland carbon sink accounted for 68.17% of the total carbon sink. This result shows that the carbon sink was proportional to the forestland carbon sink, and the forestland was the main source of the carbon sink. In 2030, the carbon sink of land use under NDS will be 560,100 tons, accounting for the total carbon sink, which is an increase of 0.1300 tons compared with 2020. Land use carbon sinks under SDS in 2030 will be 57,800 tons, an increase of 19,200 tons compared to 2020. The forestland under the SDS will be larger than that under the natural development scenario, which provides more carbon sinks. The carbon emission of urban land under the NDS will be 7.2244 million tons more than that under the SDS. This result shows that the urban land area under the SDS will be less than that under the NDS. The net carbon emission intensity of land use showed a decreasing trend, from 10,900 tons/100 million yuan in 2000 to 70 million tons/100 million yuan in 2020, a decrease of 35.78%. The net carbon emission intensity of land use under the NDS was 38 million tons/100 million yuan, and the net carbon emission intensity of land use under the SDS was 37 million tons/100 million yuan. The land use net carbon emission intensity under the NDS was higher than that under the SDS.

**Table 6.** LUCE in Shandong Province from 2000 to 2030 (Unit of LUCE: 10,000 tons, unit of net LUCE intensity: 10,000 tons/100 million yuan).

| Year | Carbon Emissions | | Carbon Sink | | | | Net Carbon Emissions | Net Carbon Emissions Intensity |
|------|----------|------------|------------|-----------|------------|-----------|----------|----------|
| | Cropland | Urban Land | Forestland | Grassland | Water Area | Bare Land | | |
| 2000 | 438.41 | 9007.57 | −32.85 | −7.01 | −8.38 | −0.11 | 9055.91 | 1.09 |
| 2010 | 416.81 | 36,541.97 | −32.33 | −5.08 | −12.58 | −0.09 | 36,592.05 | 1.08 |
| 2020 | 395.78 | 49,013.90 | −38.09 | −4.09 | −13.68 | −0.02 | 49,353.80 | 0.70 |
| 2030 nature | 372.35 | 61,121.33 | −41.32 | −3.12 | −11.54 | −0.02 | 61,437.67 | 0.38 |
| 2030 sustainable | 372.35 | 60,398.89 | −42.36 | −3.12 | −12.30 | −0.02 | 60,713.44 | 0.37 |

*3.4. Classification of LUCE Risk Areas*

Based on the regional risk indexes obtained by Formula (16), the LUCE risk was divided into five levels by the natural break point method, and the results were [0.07,0.26), [0.26,0.31), [0.31,0.43), [0.43,0.69), and [0.69,1.50], where [0.07,0.23) is a no-risk area, [0.26,0.31) is a mild-risk area, [0.31,0.43) is a moderate-risk area, [0.43,0.69) is a high-risk area, and [0.69,1.50] is a severe-risk area.

Figure 8 shows the risk level of LUCE in Shandong Province. In 2000, the whole province was dominated by no-risk areas, with a small number of moderate and mild-risk areas scattered, mainly in northern Shandong and a small number in the eastern coastal area. In 2010, it was dominated by no-risk free and mild-risk areas, of which the risk grade change areas were mostly distributed in western Shandong, southern Shandong, and eastern Shandong, and a few high-risk areas were sporadically distributed in eastern Shandong and central Shandong. In 2020, mild-risk and moderate-risk areas were the main areas. The regions with relatively large risk changes were mainly distributed in the south, northwest, north, and southeast. The no-risk areas were mainly distributed in the central and southern parts and the eastern part of Shandong Province. In 2030, most parts of the province will be dominated by mild-risk areas and moderate-risk areas. Compared with 2020, moderate-risk areas and high-risk areas will increase significantly. Under the SDS, the area of no-risk and mild-risk areas was larger than that of NDS, and the area of moderate and high-risk areas was smaller than that of NDS. This result shows that the SDS has more advantages than the NDS.

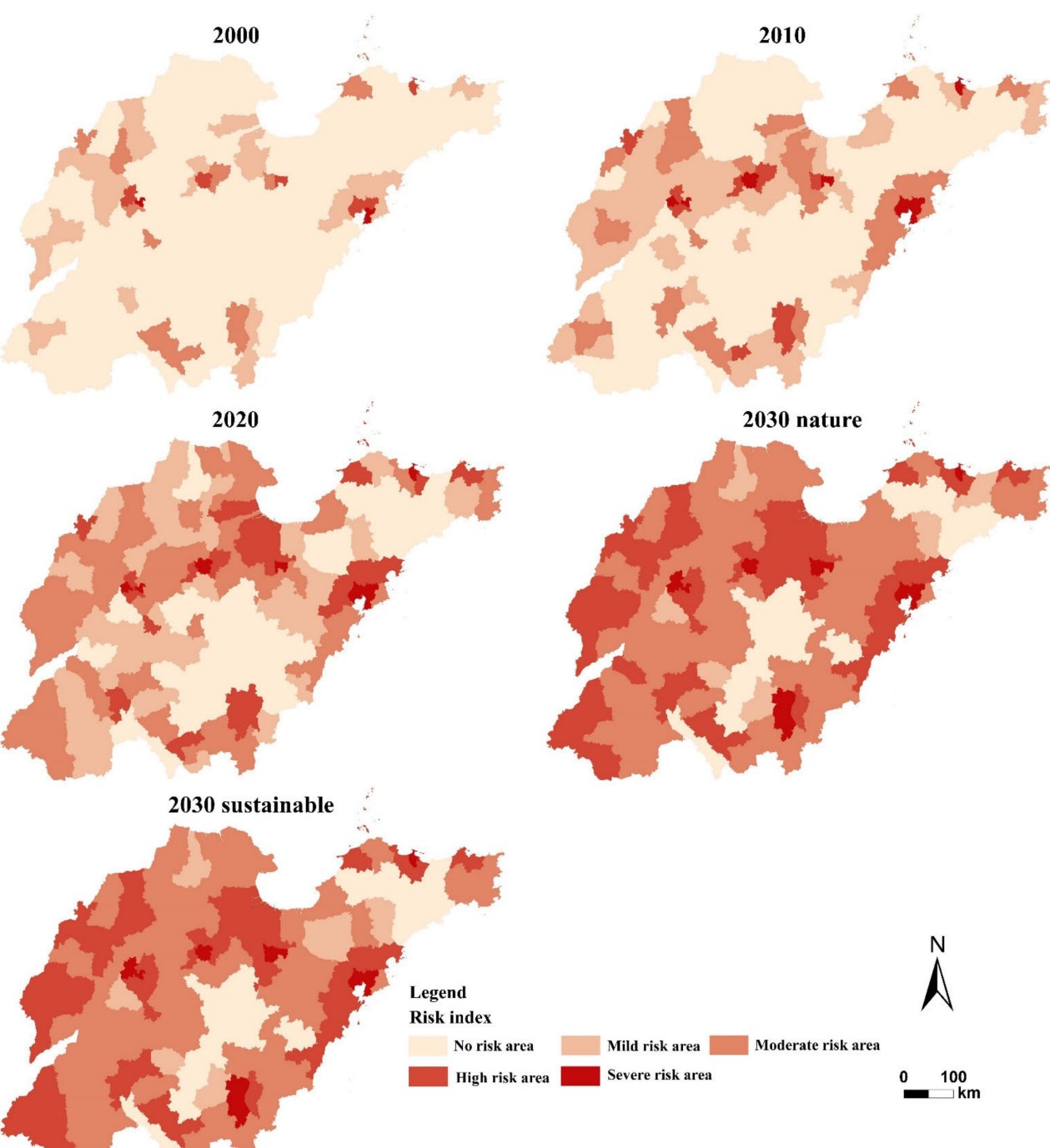

**Figure 8.** Risk level distribution map of carbon emissions from land use in Shandong Province from 2000 to 2030.

## 4. Discussion

Previous studies have predicted carbon emissions from the perspective of system dynamics [33] and economic population [34]. This method is suitable for large regions from the perspective of estimating total carbon emissions, but it is difficult to be effective for small regions. From the perspective of land use, this method can estimate the carbon emissions according to the land use of different regions. This method can be used to estimate large regions and small regions and can identify the LUCE risks of different regions. We simulated the land use of Shandong Province, a strong economic province in China in 2030

through the PLUS model. The LUCE coefficient of urban land in 2030 was predicted by using the gray forecasting model, the carbon emissions under NDS and SDS were obtained, and the risk levels of different regions were divided. This result has positive implications for policymakers formulating low-carbon emission reduction policies. Compared with other models, the prediction accuracy and confidence level of the PLUS model used in this study were higher. When studying the ecological zoning planning and dynamic assessment of Bortala Mongolian Autonomous Prefecture, Wang et al. [35] compared the prediction accuracy of the PLUS model and the FLUS model. The results showed that the PLUS model had a higher simulation accuracy and more reliable simulation results. Models such as the CA-Markov model, CLUE-S model, and FLUS model can achieve high accuracy in small-scale research, but the results are poor in large-scale research. The PLUS model is based on solving this problem [36]. Therefore, we selected the PLUS model to apply it in Shandong Province, and the research results have high credibility.

There are limitations in this study. First of all, the accuracy of the data will affect the simulation accuracy of the PLUS model, and the use of higher spatial resolution data will achieve more accurate results. Secondly, in the process of using PLUS to simulate LUCC in 2030, the rules for the NDS and the SDS are relatively simple, and the impact of the current economic development policy on land type conversion was not considered. Finally, we divided the land types into six types, namely cropland, forestland, grassland, water area, bare land, and urban land. The land types were not further subdivided; therefore, the experimental plan needs to be further improved. For future research, a more accurate method will be used to obtain more accurate LUCE measurement results.

## 5. Conclusions

This study analyzed the LUCC in Shandong Province from 2000 to 2020 and simulated the LUCC in Shandong Province in 2030 under NDS and SDS using the PLUS model. The LUCE and LUCE risk index in Shandong Province from 2000 to 2030 were measured from the perspective of land use, and the LUCE risk zones were delineated. The conclusions of this study are as follows:

(1) From 2000 to 2020, the area of cropland decreased and the area of urban land increased. Among them, the area of urban land increased by 11,616.39 km$^2$, an increase of 50.75% compared to the area in 2000, mainly due to a large amount of cropland converted into urban land. The net carbon emissions intensity decreased, meaning that each additional unit of GDP produced fewer carbon emissions;

(2) LUCE is proportional to urban land area: the larger the urban area, the more carbon emissions are generated. The LUCE generated under SDS was less than that under NDS, and rational interventions in land use patterns can reduce carbon emissions;

(3) The SDS had fewer moderate-risk areas and fewer high-risk areas than the NDS. From 2000 to 2020, the LUCE risk areas in Shandong Province were mainly no-risk and mild-risk areas, but there was a tendency for the LUCE risk level to increase over time. The high-risk areas and severe-risk areas were mainly located in economically developed central urban areas.

Promoting the optimization of land use types in terms of quantitative structure and spatial layout is of great significance for the reduction in carbon emissions. However, no further research has been performed in this paper on how to adjust the land use structure and spatial layout. In future research, it may be possible to find the best land use pattern that can effectively reduce carbon emissions.

**Author Contributions:** Conceptualization, W.Q.; Data curation, J.-L.Z.; Investigation, J.-L.Z.; Methodology, J.-L.Z., M.-H.X. and Y.-X.X.; Project administration, W.Q. and M.D.; Resources, W.Q.; Software, J.-L.Z. and J.-Y.Z.; Supervision, W.Q.; Validation, J.-L.Z.; Visualization, J.-Y.Z. and W.Q.; Writing—original draft, J.-L.Z.; Writing—review and editing, J.-L.Z., W.Q. and Z.-J.Z. All authors have read and agreed to the published version of the manuscript.

**Funding:** This research was funded by the Remote sensing monitoring of cultivated land and establishment of its benchmark land prices, grant number 381180/010, and Open fund of Key Laboratory of Land Surface Pattern and Simulation, Chinese Academy of Sciences, grant number LBKF201802.

**Institutional Review Board Statement:** Not applicable.

**Informed Consent Statement:** Not applicable.

**Data Availability Statement:** Related data are available upon reasonable request.

**Conflicts of Interest:** The authors declare no conflict of interest.

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
