# Peer review of "Land Use Carbon Emission Measurement and Risk Zoning under the Background of the Carbon Peak: A Case Study of Shandong Province, China"

_sustainability, doi:10.3390/su142215130_

Round 1

Reviewer 1 Report (Previous Reviewer 1)

I sent my comments to the editors.

Author Response

Reviewer 2 Report (Previous Reviewer 3)

Revisions fulfil my comments about the manuscript.

Author Response

Thank you again for your valuable suggestions to make the manuscript perfect and best wishes.

Reviewer 3 Report (Previous Reviewer 4)

Dear authors,

Thank you for the revision of the manuscript. I only have a few small recommendations left.

1. It is necessary to improve the readability (increasing the font) in Figure 2 and in the legend in Figure 7.

2. Please, do not use the same words and phrases in the Keywords as in the title of the manuscript. This reduces the effectiveness of their use. Replace them.

3. Improvement of the English language of the manuscript is still needed. For example, the phrase "Land use cover change (LUCC)" (line 43) is incorrect.

Round 2

Reviewer 1 Report (Previous Reviewer 1)

-

Author Response

Thank you again for your helpful suggestions to improve our manuscript. 

This manuscript is a resubmission of an earlier submission. The following is a list of the peer review reports and author responses from that submission.

Round 1

Reviewer 1 Report

In this manuscript, emissions of land uses are measured, and then the risk zones are prepared using PLUS and Gray models. I did not find scientific flaws in the manuscript, and it seems that the models applied in this study were correctly connected. Although the area of land-use emission evaluation is noteworthy per se, the contribution of this research to the existing literature is minor. Prediction of the future land uses in the Shandong area can be deemed an innovation, but it is not a novel idea presented for the first time. In other words, coupling several models (PLUS and Gray methods in this case) is not necessarily lead to a new strand of research. In addition, the findings are case-specific and do not assist the scholars in the land-use and environmental planning community in perceiving the land-use emission phenomenon as a whole notion more accurately.

Reviewer 2 Report

The authors submitted a well written and an interesting manuscript. How the manuscript should be revised, especially the section of introduction and methodology, before it could be considered for publication. Below are some comments and suggestions to improve the manuscript:

Lines 39-46: The references [2-5] provided are very old, it could be better to refer to very recent studies.

Lines 33-79: In the section of Introduction, the authors should provide an introduction or a brief review of the studies which have been conducted using their proposed model “PLUS model and the Gray model” and specify their innovation or contribution in the application of these models for the prediction of the carbon emissions.

Lines 103-107: The description of the data sources is not sufficient. Please provide more details other than the ones provided on Table 1. In addition, there are no dates on which the links provided have been accessed to.

Line 109: Please provide the references.

Lines 201-204: Since we do not have enough description indicating accuracy and precision of the real 2020 land use data, it could be difficult to evaluate the results of precision accuracy assessment of the proposed models.

Lines 126-211: The references provided in the section of Methodological framework are not accessible and some of them are not in English language. It would be hard to verify the statements quoted from these references. Please provide recent, open accessible and English redacted references. 

Reviewer 3 Report

The research calculates the carbon emission of land use in Shandong Province in 2030 under natural development scenario and sustainable development scenario by constructing PLUS model and gray model. It deals with an important topic which is the estimation of carbon emission in the future based on land use change.

Main concern:

Line 73-75: “This study uses the combination of the PLUS model and the Gray model to predict

the carbon emissions of Shandong Province in 2030, providing a new method for carbon

emissions prediction.”: I think no any combination and new method is provided in the methodology and results sections.

Line 168: “introducing the gray model GM (1,1)”: this model is not new how do you introduce it? It is just an application.

Minor concern:

Line 29: “the high-risk zone and the high-risk zone”. Correct it.

Line 37” “,To”

Line 38-40 and the entire manuscript: You need to add one space with the starting of a new sentence.

In the abstract: “the carbon emission per unit of GDP decreases gradually” but in the study area you mentioned: “While the regional GDP increases year by year, carbon emissions also increase year by year.”

 Line 113: “plus model”, the name of the model should be in capital letters.

Reviewer 4 Report

I believe that the land-use carbon emission measurement presented in the manuscript is one-sided. Although land-use/-cover changes estimates by 2030 are possibly close to reality, these changes are occurring and will continue to occur in the coming decades against the backdrop of rapidly growing climate change. In the dynamics of carbon, ecosystems are sensitive to changes in air temperature. This primarily concerns the boundaries between the soil and atmosphere, the water surface and the atmosphere. The role of moisture/humidity (expressed in terms of some coefficient of moisture) can also be important, for example, in the activation of microbiological processes in the soil, etc. However, the authors did not take into account the role of potential climate change in the forecast of carbon emission between 2020 and 2030 (within each land-use/-cover unit) in the studied province of China. Then, what is the scientific and practical value of the findings? In my opinion, it is only in the results of forecasting the dynamics of land-use change, and not carbon emission changes.

In addition: In the manuscript, there is no information about the important limitations and uncertainties of the study. Therefore, the results of the work are not self-critical.

As it stands, I cannot recommend the manuscript for publication.